# Spark of Life: Role of Electrotrophy in the Emergence of Life

**DOI:** 10.3390/life13020356

**Published:** 2023-01-28

**Authors:** Guillaume Pillot, Óscar Santiago, Sven Kerzenmacher, Pierre-Pol Liebgott

**Affiliations:** 1Center for Environmental Research and Sustainable Technology (UFT), University of Bremen, 28359 Bremen, Germany; 2Aix Marseille University, Université de Toulon, IRD, CNRS, MIO UM 110, 13288 Marseille, France

**Keywords:** emergence of life, hydrothermal vents, electrotrophy, prebiotic synthesis, electroreduction of CO_2_

## Abstract

The emergence of life has been a subject of intensive research for decades. Different approaches and different environmental “cradles” have been studied, from space to the deep sea. Since the recent discovery of a natural electrical current through deep-sea hydrothermal vents, a new energy source is considered for the transition from inorganic to organic. This energy source (electron donor) is used by modern microorganisms via a new trophic type, called electrotrophy. In this review, we draw a parallel between this metabolism and a new theory for the emergence of life based on this electrical electron flow. Each step of the creation of life is revised in the new light of this prebiotic electrochemical context, going from the evaluation of similar electrical current during the Hadean, the CO_2_ electroreduction into a prebiotic primordial soup, the production of proto-membranes, the energetic system inspired of the nitrate reduction, the proton gradient, and the transition to a planktonic proto-cell. Finally, this theory is compared to the two other theories in hydrothermal context to assess its relevance and overcome the limitations of each. Many critical factors that were limiting each theory can be overcome given the effect of electrochemical reactions and the environmental changes produced.

## 1. Introduction

Whether it begins elsewhere in the Universe or on Earth, many hypotheses have been established about the origin of Life throughout history. Starting with divine creation and then spontaneous generation, scientists have arrived at the hypothesis of the prebiotic transformation of inorganic matter into organic matter, which would have self-organized in the course of time to form modern complex living organisms. In this last theory, different approaches have been used to try to define the conditions of the emergence of Life. Whereas geochemists attempt to understand the transition from inorganic to organic [1,2], biologists attempt to trace the evolution of modern species back to the “Last Universal Common Ancestor” (LUCA) [3,4], the theoretical phylogenetic ancestor of all life on Earth. At an intermediate scale, other researchers focused on different functional aspects that define Life, such as the self-replication and diversification of molecules as in the ‘RNA world’ [5,6]. Here, we aim to draw a parallel between a newly discovered modern metabolism, named electrotrophy, and the first step of the emergence of Life, with the transition from inorganic to organic, to go further down the road to a new theory for the potential development of each mechanisms leading to the first living organism.

The recent discovery of putative microfossils in hydrothermal spring precipitates dated between 3.77 and 4.28 billion years ago, the oldest found at present, have provided a likely historical context for the emergence of Life [7]. Thus, the first living organisms would have appeared during the Hadean, more than 3.8 billion years ago, after the formation of the oceans (4.40–4.26 Ga) and the stabilization of the early Earth. Thus, in order to study the potential mechanisms for the emergence of Life on Earth, it is necessary to look at the conditions of the primitive Earth to define the most favorable environment for the development of the first biological elements and proto-cells.

## 2. The Cradle of Life

As much as modern living organisms are complex and sensitive to their environment, the cradle of Life would require a combination of specific environmental conditions, allowing the formation of organic compounds, their stabilization, and condensation into bigger molecules. Researchers have looked for these niches [8,9,10,11,12,13,14] and identified several hotspots, such as ponds, hot springs, aerosols, asteroids, beaches, lagoons, and more based on different aspects. The most popular, the hydrothermal systems, present a number of essential conditions for the emergence of Life. Indeed, such conditions could be water-based environments, steep gradients of redox conditions, of pH, of temperature, and of the presence of metallic catalyzers, and the protection from UV radiation from the surface. The energy source has been hypothesized to be first the sulfur compounds of acidic systems, in the “Iron-Sulfur World” theory [9]. Due to the too high instability of organic molecules in the presence of H_2_S present in the hydrothermal fluid, this hypothesis has lost popularity. With the discovery of alkaline vents in 1979 [15], a new theory emerged, considering first H_2_ and then CH_4_ as an energy source (electron donor), in the Alkaline Hydrothermal Vent (AHV) theory [10,16,17,18,19]. These hydrothermal vents are fueled by the hydration of ultramafic rocks, known as the serpentinization reaction (Peridotite + H_2_O → Serpentinite + H_2_ + CH_4_ + heat), which produces alkaline hydrothermal fluid. However, with recent and future discoveries, other energy sources could be considered for the establishment of new theories, such as spontaneous and natural electrical currents found in the environment.

## 3. Natural Electrical Current in Hydrothermal Chimneys

During the last decade, a Japanese research team has demonstrated the existence of an electric current within acidic or mixed hydrothermal vents [20,21,22]. The walls of these chimneys are made of polymetallic sulfides (pyrite (FeS_2_) or chalcopyrite (CuFeS_2_)), which is electrically conductive or semi-conducting, with resistivities ranging from 0.1 to 105 Ω·m [23], whereas the typical range of resistivity for metals is from 10^−8^ Ω·m to 10^−6^ Ω·m and insulators from 10^8^ to 10^18^ Ω·cm. These minerals are therefore capable of conducting electrons over distances of several centimeters between environments at different redox potentials [20]. The wall of hydrothermal vents can then be schematized as a porous and conductive membrane that separates two compartments with radically different physicochemical conditions. Thus, the difference in potential between the reduced fluid rich in H_2_S (estimated at −0.4 V vs. SHE, with 0.1 M Na_2_S, pH 12, 30 °C) and the oxygenated seawater (estimated at +0.4 V vs. SHE, in O_2_-purged aqueous solution of NaCl at 0.17 M) leads to an electromotive force across the wall of the chimneys that allows the production of a continuous electric current. This electric current has been measured in situ and reproduced in the laboratory [21,22] by mimicking the conditions found within acidic hydrothermal vents, producing up to 200 mW·m^−2^ in the laboratory [22].

This discovery raises many questions about the role of this electric current in the origin of Life in a hydrothermal context. Indeed, everything suggests that the conditions for the formation of this electric current were also present during the Hadean [24]. The hydrothermal fluids of the Hadean were also composed of hydrogen sulfide and other reduced compounds, producing hydrothermal vents similar in structure to modern vents. The absence of oxygen in the seawater was probably compensated by the presence of oxidized molecules such as nitrate, iron, or sulfate that act as electron acceptor in these electrochemical reactions. In fact, previous work has suggested that nitrogen oxides, such as nitrate, would have been produced by atmospheric photochemical reactions, discharging significant amounts of NO_3_^−^ into the oceans, accumulating over millennia [25]. Concentrations up to 24 mM were estimated depending on the atmosphere composition in CO_2_ [25]. These concentrations were contested by a later publication and lowered to the sub-micromolar range due to the destruction from UV, hydrothermal vent conditions, and soluble Fe^2+^ [26]_._ Considering the estimated environmental conditions from the literature for the emergence of Life, the onset potential of the different electrochemical reactions can be calculated using the Nernst equation, considering the dependence of the standard potential with the temperature [27]. According to these thermodynamic calculations, nitrate reduction into ammonium is a promising candidate to replace O_2_ reduction as it presents a similar onset potential around 0.38 V vs. SHE (Figure 1) at a sub-micromolar level, with an estimated potential of −0.39 V vs. SHE for H_2_S oxidation into solid sulfur at the “anodic” side of the chimney wall. The coupling of these reactions leads to a Gibbs free energy of −596 kJ/mol, threefold higher than the coupling H_2_S oxidation with O_2_ reduction with −157 kJ/mol (Figure 1, scenario 2 and 1). This overall voltage can even be higher when we consider the oxidation of H_2_, produced by serpentinization reactions such as in the AHV [28], in an alkaline condition, with a potential estimated at −1.02 V vs. SHE, which leads to a free Gibbs energy of −1085 kJ/mol when coupled with nitrate reduction (Figure 1, scenario 3). However, the most recently estimated low concentration of nitrate might considerably reduce the kinetics of the reactions. Further investigation might give a more precise view of nitrate concentration in Hadean oceans and its associated role in the electrical current production. Other electron acceptors, with more negative onset potentials could also be an alternative, such as CO_2_ (see Section 5) or Fe^2+^. Thus, from a thermodynamic point of view, this energy source could have played an important role in the formation of organic molecules by supplying electrons to reduction reactions taking place at the surface of hydrothermal vents.

## 4. Electrotrophy in Modern Hydrothermal Vents

In recent studies, the ability of some microorganisms from modern hydrothermal vents to grow using only electricity as an energy source has been demonstrated, either focusing on hyperthermophiles [35,36,37], or in situ, using electrical current generated from a hydrothermal fluid [38]. This metabolism was defined as a new trophic type, as compared to photo- or chemo-trophy, named electrotrophy [39]. In the studies focusing on hyperthermophiles, the enrichment of electro-auto-trophic communities was performed in Microbial Electrochemical Systems (MES) at 80 °C, with a polarized electrode as the only energy source, CO_2_ as the only carbon source, and different electron acceptors (nitrate, sulfate, oxygen, and iron (III) oxide). The cathode was first polarized at −590 mV vs. SHE to ensure providing enough reducing power to drive the electro-auto-trophic metabolisms and obtain dense biofilms. The cathode was then polarized at −300 mV vs. SHE, a more positive potential where H_2_ evolution is thermodynamically not possible at 80 °C, to check the independence to H_2_ mediated mechanism. With each electron acceptor and cathode potential, the development of a consequent biofilm was observed on the cathode allowing to affirm the enrichment of electrotrophs from a conductive support as the only energy source. The conditions used, mimicking those found in hydrothermal vents (potential of the conductive surface, pH, temperature, nutrient composition), led us to hypothesize that such metabolisms are directly used by some microorganisms found in these ecosystems, which was recently supported with the in situ enrichment of electrosynthetic communities [38].

The understanding of the biodiversity of electrotrophs remains limited, as more and more electrotrophs are discovered over time. Enrichment from hydrothermal vents at 80 °C, previously mentioned, has shown the systematic presence of Archaea from the taxonomic groups of Archaeoglobales and Thermococcales [35]. The study of pure cultures of 11 hyperthermophilic strains, initially isolated from deep-sea hydrothermal vents, have confirmed their ability to grow on cathodes and produce relatively high amounts of organic acids [37]. The in situ enrichment showed the specific growth of *Candidatus Thiomicrohadbus electrophagus* on carbon felt sheet compared to the insulated control [38]. Other electrotrophic microorganisms from different phylogenetic groups have been identified over the last decade, such as *Kyrpidia spormannii* (Bacillota) [40], *Acidithiobacillus ferrooxidans* (Acidithiobacillia) [41], *Mariprofundus ferrooxydans* (Zetaproteobacteria) [42], some methanogens (*Methanothrix* spp. [43], *Methanosarcina* spp. [44], *Methanobacterium* spp. [45]), and some acetogens [46]. Many of these strains have been isolated from hydrothermal systems or can be found in those environments. However, by definition, electrotrophy applies to all electron transfer from a conductive solid particle, such as iron [39]. Then Iron-Oxidizing Bacteria using solid Fe(0) or Fe(II) as electron donor are also electrotrophs lengthening the list of electrotrophs. Thus, it can be concluded that this metabolism is widely spread in the tree of Life and could potentially have a common origin from LUCA.

The extracellular electron transfer (EET) mechanisms involved in electrotrophy are still poorly understood and the current state of the art has already been recently covered [39]. The main actors identified so far are periplasmic multiheme c-type cytochromes. Other potential mechanisms were hypothesized, such as the use of conductive nanowires or the use of a redox mediator (flavins, H_2_, etc.) shuttling electrons between conductive support and cell. Although a recent redefinition of electrotrophy excludes the use of mediators, such as the instance when H_2_ is produced by water electrolysis, if the oxidation process occurs intracellularly, we consider here the importance of this mechanism for the emergence of Life, as it can provide energy at the range of the nanoscale. Indeed, considering the effect of the electromotive force of the steep gradients previously mentioned, the electrical current could produce undetectable amounts of H_2_ in hydrothermal systems (onset potential of H_2_ oxidation tending to 0 V vs. SHE at pH 3, 25 °C when H_2_ concentration is in the nanomolar range), especially when H_2_ is absent from the hydrothermal fluid, allowing the growth of hydrogenothrophs with high affinity to H_2_. The same reasoning is also true for prebiotic syntheses, which allows reactions normally reserved to H_2_-producing alkaline vents, such as in the AHV theory, and in other acidic hydrothermal systems where an electrical current is present.

## 5. Prebiotic Synthesis and CO_2_ Electroreduction

In all hypotheses of the origin of Life, it is necessary to focus on the transition from inorganic to organic, and the separation of the proto-cell from the environment. These prebiotic syntheses have been studied for decades, but remain a limiting step in understanding the emergence of Life [47]. Since 1988, the work of Wächtershäuser and his successors on the Iron–Sulfur World has been focused in studying the mechanisms of prebiotic synthesis in an acidic or sulfuric hydrothermal context (Figure 2), highlighting different reactions of polymerization of organic matter, without ever succeeding in showing the reduction of CO_2_ in realistic conditions possible in the nature [9]. In the AHV theory, the serpentinization reaction, at the origin of the alkaline vent, produces H_2_ in great quantity which can be transformed into short carbon fatty acids (SCFA) and amphiphilic fatty acids by processes of Fischer–Tropsch types (Figure 2) and other non-enzymatic reactions [48].

Now, with the discovery of this electrical current, electrochemical reduction of CO_2_ is also to be considered more closely (Figure 2). When looking at the standard potentials at 25 °C of CO_2_ reduction (from −0.15 to +0.16 V vs. SHE, depending on the product of the reaction) compared to the anodic reactions at the same temperature (+0.144 for H_2_S oxidation and −0.091 V vs. SHE for H_2_ oxidation), these reactions appear immediately non-spontaneous. However, a previous study has shown that when considering the proton (pH) and thermal gradients in hydrothermal conditions, these reactions become more favorable, with potential of −0.78 V vs. SHE for H_2_ oxidation and −0.43 V vs. SHE for reduction of CO_2_ to CO [49]. In complement of the proton and thermal gradient, the concentration gradient of each component of the reactions influences its onset potential, leading to an even larger potential difference with a rough estimation of −1.02 V vs. SHE for H_2_ oxidation and −0.29 V vs. SHE for CO_2_ reduction into CH_4_; for example, when considering already reported concentration in deep-sea or hydrothermal systems (Figure 1). The Gibbs free energy of such reaction would be around −470 kJ/mol, demonstrating its spontaneity. In 2014, a study demonstrated that the electroreduction of CO_2_ to CO and CH_4_ in FeNi_2_S_4_ (violarite) hydrothermal deposits coupled with the oxidation of H2 was spontaneous under slightly acidic conditions [50]. Another recent study demonstrated that the abiotic electrical current within hydrothermal vents could lead to the reduction of CO_2_ to CO and formate on metal sulfide crystals (CdS and CuS), typically found in these environments [51].

The effect of the distance between the chimney wall and the catalytic cathodic site on the CO_2_ reduction kinetics has also been reviewed [49]. Indeed, the gradients of pH, redox potential and temperature along the chimney wall produce a diversity of catalytic points with different conditions and different proton, thermal, and concentration motive-forces, influencing the onset potentials of each reaction. This finding fuels hypotheses of the abiotic production of organic molecules condensed on the surface of hydrothermal vents by CO_2_ fixation in a primordial soup, and provides the building blocks for the formation of a living organism.

One fundamental aspect of Life is the compartmentalization of the inside versus its environment, allowing the concentration of compounds and the production of an intracellular microenvironment more thermodynamically favorable for all the biological functions, such as polymerization, protein folding, etc. Studies have shown that the prebiotic syntheses through the aforementioned Fischer–Tropsch and Wächtershäuser reactions could have allowed the production of amphiphilic molecules (lipids from C12 to >C33 at 150–250 °C), which gathered in micelles to constitute the proto-membranes of the first cells [52,53,54] (Figure 2). The morphology of these micelles differs, depending on the fatty acid composition, the salt concentration and the pH, with mainly a monolayer at low pH, and a bilayer at high pH. These micelles also have the property to slow down the diffusion of molecules and, by the action of the mechanisms of organic production with fast kinetics, allow for the concentration of compounds inside, which produces a redox gradient with the outside. These micelles also tend to adhere to the surfaces, creating a microscale environment on the chimney surfaces that could be fed by the electrical current. Although the synthesis of micelles from CO_2_ electroreduction have not been reported yet, we believe that such process could also be possible using the right conditions and catalysts (see Section 8). These proto-membranes would then be stabilized over time by the addition of more complex molecules, such as glycerol-ether and glycerol-ester lipids, constituting the modern cell membrane of Archaea and Bacteria, respectively [55].

## 6. Energetic Proto-Metabolism

Once the proto-cell is separated from the environment and can concentrate organic compounds, it is necessary to look at the development of a proto-metabolism. In 1944, Erwin Schrödinger had already mentioned the necessity of a constant energy supply from the environment for the synthesis of biological material and the establishment of biological functions for the emergence of Life [56]. Numerous studies have focused on the potential electron donors available in sufficient quantities in the deep hydrothermal springs of the primitive Earth.

Based on modern chemosynthesis, three main molecules were identified as sufficiently energetic (low potential) to serve as energy sources for proto-metabolism: H_2_, CH_4_, and H_2_S [56]. Subsequently, H_2_S was considered too reactive to allow the synthesis and maturation of organic molecules. For a decade, H_2_ was considered a good candidate, giving rise to hypotheses of the origin of Life from the serpentinization phenomena of alkaline smokers very rich in H_2_ [10,16]. This H_2_ can then react with CO_2_ through Fischer-Tropsch–type reactions, producing hydrocarbons and CH_4_ which can also serve as a source of both electrons and carbon for the proto-metabolism [17,57,58,59]. Thus, the debate continues in the scientific community about the most likely energy source for the emergence of proto-metabolism.

With the recent discovery of electrical currents within deep-sea hydrothermal vents [20], a new potential energy source has been identified. As previously mentioned, recent studies have demonstrated the ability of modern electrotrophs from hydrothermal vents to grow from an external electric current, coupled with the fixation of CO_2_ and the reduction of different electron acceptor, such as nitrate, sulfate, CO_2_, O_2,_ and iron-oxide [35,36,37]. Based on these observations, we can assume that Life has evolved to use an electrical current in combination with a range of different electron acceptors. It is therefore necessary to look at the most probable one present during the emergence of Life.

For some researchers, nitrate, produced by atmospheric photochemical reactions, is considered the most likely electron acceptor at the origin of Life on Earth [60,61]. It was probably the most oxidized molecule present and could have replaced oxygen as the electron acceptor for the production of the abiotic current, as previously stated (Figure 3). In fact, a study has shown the natural reduction at 120 °C under acidic conditions of nitrate to ammonium by oxidation of pyrite particles, which constitute the deep hydrothermal chimneys [62]. The addition of an electrical current would have even eased and fasten this reaction on conductive pyrite by feeding the reaction with lower potentials and electron supply. Moreover, we can suppose that different reactions could have occurred at the same time, such as the CO_2_ reduction previously mentioned, depending on the onset potential of these reactions. A screening of all the possible reactions and their onset potential and specific overpotential on hydrothermal chimney material in such an environment could help predict the possible components reacting and concentrating on the surface of the chimney.

Then if we hypothesize a proto-metabolism using an electrical current as the electron donor and nitrate as the electron acceptor, it is necessary to look at the modern electrotrophic pathway and its plausibility in terms of evolution and phylogeny. The main key enzymes involved in these pathways are cytochromes, ferredoxin, and nitrate reductase, which were supposedly present in LUCA [63,64], supporting this hypothesis. These enzymes would have been complex and already the product of millennia of evolution. A simpler version would be considered at the emergence of Life. As discussed earlier, proto-membranes would most likely have formed by the association of micelles of amphiphilic molecules on the surface of hydrothermal vents [52]. Metallic catalysts constitute the core of the modern hydrothermal chimney, with up to 60 wt% of Fe, 33 wt% of Cu, 53 wt% of S, 3 wt% of Ni, and traces of Ag, Zn, Cd, or Co, in the constituting minerals [65]. The incorporation of metallic nanoparticles, of the FeS or NiFe type for example, into the proto-membranes would have allowed the conduction of electrons from the chimney to the interior of these proto-cells. These incorporations of conductive metallic nanocrystals in the membranes can be compared to the active sites of the modern metallo-enzymes, such as cytochromes or nitrate reductase [39].

Thus, the compartmentalization of the cathodic reaction (where the electroreduction in compounds is happening) in this proto-membrane would have been the first step for the establishment of a proto-metabolism. However, in modern respiring organisms, the energy produced by the reduction of an electron acceptor is used to transfer protons outside the cytoplasmic membrane. This creates a proton-motive force that feeds ATP synthase and ultimately leads to the production of ATP, a form of intracellular energy storage and transport. However, this system appears already complex, and a more archaic system probably initiated this proton-motive force. It is therefore necessary to look at proton transfer in electrotrophy and in an electrochemical context.

## 7. Proton Gradient

It is well known that in all electrochemical systems an ionic transfer takes place between the anode and the cathode of the system through the electrolyte, in parallel to the electron transfer through the external circuit. The use of ionic membranes to separate the two electrodes can slow this ionic transport and lead to the creation of a pH difference between the two compartments. Indeed, the anode produces protons during oxidation, whereas the cathode consumes these protons during reduction. If the ionic transport is slowed down, protons accumulate at the anode, decreasing the pH of the anode compartment, while the pH of the cathode compartment increases by consumption of protons at the cathode.

The proto-membrane of the proto-cell can be assimilated to the ionic membrane separating the anode and the cathode. The anode is represented by the inner wall of the chimney where the H_2_S or H_2_ oxidation reaction takes place (Reaction 1, Figure 3), whereas the cathode is represented by the outer wall of the chimney where the proto-cell-mediated nitrate reduction takes place (Reaction 2, Figure 3). In this case, the reduction of nitrate, or other compounds in the proto-cell, would lead to the “intracellular” consumption of protons, with 10 moles of proton consumed per mol of nitrate, for example. However, the semi-permeable micellar membrane could limit the entry of extracellular protons, allowing the creation of an H^+^ concentration gradient. The result of this concentration gradient and electrical potential (incorporation of conductive metallic nanocrystals) would have constituted the electrochemical gradient allowing the establishment of a proton driving force (Reaction 3, Figure 3) similar to the one observed in modern microorganisms. The transition from proton gradient to ATP synthase remains obscure as no intermediate state have been found yet giving clues on the possible evolution mechanism. Therefore, the modern out-of-cell proton transport system (respiratory chain complex) required for the creation of the driving proton force would potentially have arisen after the establishment of the primitive ATP synthase system, as hypothesized in the origin of Life hypotheses in alkaline vents [66] (Figure 4). The ATP synthase system would likely have been built around the natural proton flux produced by the delocalization of redox reactions from proto-metabolism through the geo-electrochemical phenomenon.

## 8. From CO_2_ Electroreduction to Autotrophic Pathway

### 8.1. Autotrophic Pathway of Electrotrophs

Once an energetic system has been set up within the proto-cell, it is now necessary to focus on the proto-system of CO_2_ fixation into more complex organic molecules. Most electrotrophs enriched on cathodes fix CO_2_ through the reductive acetyl-CoA, also called the Wood–Ljungdhal (WL) autotrophic pathway [36,46]. This autotrophic pathway is arguably considered to be the oldest pathway, whose genes were present in LUCA [67] because it is the least energy intensive (1 mole of ATP per mol of CO_2_). Thus, a similar primitive pathway could probably be the origin of CO_2_ fixation into organic matter. The modern pathway relies on metallo-enzymes and is composed of two branches: (i) the carbon branch and (ii) the methyl branch, allowing the formation of acetyl-CoA. The carbon branch is composed of a CO dehydrogenase (CODH), which transforms CO_2_ into CO, and is linked to acetyl-CoA synthase, combining CO, a coenzyme A, and a methyl to form acetyl-CoA. This branch is common to archaea and bacteria. However, the methyl branch differs between archaea (methanogens and Archaeoglobales) and bacteria (acetogens). Most steps in this pathway use reduced ferredoxins to allow the transfer of electrons necessary for the reduction of CO_2_.

### 8.2. Autotrophic Pathways at the Emergence of Life

Previous experiments have indicated that under conditions mimicking hydrothermal vents, an electric current could have a direct effect on this pathway in Archaeoglobales [36]. Indeed, the constant influx of electrons into the cell causes an overactivity of this CO_2_ assimilation pathway, through the rapid and continuous turnover of reduced ferredoxins, leading to the accumulation of pyruvate in the medium. The enzymes involved in this pathway are composed of a metallic active site (iron–sulfur proteins containing nickel for some) but remain very complex and are the result of a long evolution. However, a recent study has shown the abiotic catalysis of some of the reactions of this pathway by metals (Fe, Ni, Mo, Co, Mn) under hydrothermal conditions (Figure 5—reactions marked with 2), leading to the specific production of the different metabolic intermediates (CO, formate, and acetate) of the WL pathway [68,69]. Electrons provided to these catalysts by an electrical current would probably increase the rate of these reactions. This demonstration seems to link prebiotic syntheses with the WL pathway as present in primary colonizers of modern hydrothermal vents. In addition, pyruvate, which is mass produced in experiments on electrotrophs from deep-sea hydrothermal vents [35], appears to be a potential key component in the emergence of Life. It was suggested that during the emergence of Life, the precursors of ribonucleotides, amino acids, and lipids would come from a single source simultaneously [70]. In modern organisms, the synthesis of carbohydrates and amino acids is achieved from pyruvate resulting from glycolysis or autotrophic carbon fixation.

The recent screening of hyperthermophilic electrotrophs have also highlighted the role of the reverse tricarboxylic acid cycle (reverse TCA), another autotrophic pathway, in the synthesis of organic compounds from CO_2_ and electricity [37]. However, when looking at its capacity to fix inorganic CO_2_ to organic compounds, we can quickly realize that each step requires an already existing C4 (Succinyl-CoA) or C5 (alpha-ketoglutarate) compound to bound to CO_2_ (Figure 5). Whereas these complex carbon molecules can be easily replicated with an existing machinery, in one of the steps, the production of building blocks directly from CO_2_ appears more challenging. However, 6 of the 11 steps of the reverse TCA could be performed by different metal catalyst (Fe^0^, Zn^2+^, Cr^3+^) when supplied with the different step compounds of the cycle (Figure 5, reactions marked with a 1) [71]. We can then hypothesize that this autotrophic pathway could have set up as a secondary CO_2_ fixation pathway after production of one of its compounds by other prebiotic synthesis mechanisms.

### 8.3. Anabolism, and Further Synthesis

After the prebiotic synthesis of small organic acids, it is necessary to look at further syntheses into lipids, carbohydrates, or proteins, which are the main constituents of the modern cells. As previously mentioned, pyruvate could be a key compound due to its more reducible carbonyl group than its carboxylic entity [77]. It could be the starting point to the lipid and carbohydrate synthesis through its transformation into glyceraldehyde. So far, only the reverse reaction, from glyceraldehyde to pyruvate, could be performed at high pH by the hydrothermal–formose reaction [72] (Figure 5, reaction 4). By this same reaction, glyceraldehyde could be transformed into ribose, which is the first step toward the synthesis of DNA and RNA.

When looking at the amino acid synthesis, seven of them have been electrochemically synthesized from α-keto acids (including pyruvate) and NH_3_ or NH_2_OH using a TiO_2_ catalyst at low pH and a relatively positive potential (−320 mV vs. SHE) [74] (Figure 5, reactions 6). Whereas hydroxylamine (NH_2_OH) concentrations in the Hadean are not well understood, NH_3_/NH_4_^+^ can be electrochemically produced from nitrogen oxides, as previously mentioned. On the other hand, formamide is considered as the first building block in many prebiotic syntheses [78]. Electrochemical synthesis of formamide and acetamide have been demonstrated from mixtures of formate or acetate and nitrite at −0.4 V vs. SHE on a Cu catalyst in conditions easily retrieved in hydrothermal context [73] (Figure 5, reactions 5). This formamide and acetamide could then react with other carbon-based molecules to produce amino acids, nucleobases, and lipid precursors.

Other experiments on electrochemical reduction of CO_2_ have demonstrated the production on Cu or Ag catalyst (which can be found in the chalcopyrite of hydrothermal chimneys) of CO [75], C_2_ aldehydes, ketones, alcohols, and carboxylic acids at potential relatively negative (−0.67 to −1.18 V vs. SHE) that could partially be obtained in hydrothermal vents [76]. Indeed, as previously mentioned, spontaneous CO_2_ reduction is thermodynamically possible when in combination with H_2_ oxidation at the anodic side (Figure 1).

### 8.4. Polymerization

When all these building blocks have been synthesized, it is time to look at the production of more complex molecules obtained by polymerization and folding, etc. One of the main challenges for polymerization is the removal of water molecules from monomers, dimers, and further polymers to bind them together. This is highly affected by the water activity and the equilibrium constant of the binding reaction, which decreases for each monomer attached. Thus, the polymerization of organic compounds, such as amino acids, is supposed impossible in aqueous conditions [79]. It is then necessary to look at alternative mechanisms. Electropolymerization of amino acids requires relatively high anodic potential (<+1 V vs. SHE) [80,81], which is not coherent with the electrotrophic hypothesis and the environmental conditions. However, extensive research has been performed on the “Clay theory”. Indeed, clays, and especially the group of smectite, have the ability to synthesize and adsorb monomers in their matrix and allow their polymerization. A recent review summarizes all the research performed on this topic [82]. Different studies have shown the presence of smectite in hydrothermal vents and the related role of pyrite in the synthesis of compounds inside the clay [83,84]. Some studies have also shown that transition metals, involved in many prebiotic synthesis, enhance the adsorption of nucleotides onto clays [85]. However, most of these studies have not been undertaken in conditions analogous to hydrothermal vents and would need further investigation to validate these processes. Microfluidic approaches, such as already developed for the AHV theory [86,87], which reproduce the hydrothermal conditions with hydrothermal chimney material as a heterogeneous catalyst and clay-like matrix, fed with hydrothermal and seawater reactants, could help fill the gaps on the prebiotic synthesis pathways. If such processes are validated, the colocalization of energy source (electricity), reduction of compounds from the environment (NO_3_^−^ to NH_4_^+^, CO_2_ to organic acids), and the adsorption and concentration of products into a mix of clay and metal catalyst would most probably result in the polymerization of increasingly complex molecules.

### 8.5. Electron Donor Mediation for Planktonic Migration

The production and accumulation of increasingly complex organic molecules in a delimited space would necessarily lead to a selective evolution of abiotic processes toward stabilized mechanisms anchored in organic matrixes (Reaction 2, Figure 6). This would produce the first proto-enzymes, harboring the primal polymetallic catalytic sites. Once these catalytic sites were fixed on a robust semipermeable membrane, they could detach from the conductive chimney and continue to catalyze the oxidation of diffuse H_2_S, H_2_, or other reduced molecules, acting as electron mediator to feed the proton gradient and the proto-metabolism previously presented (Figure 4 and Figure 6). This phenomenon is already observed in modern electrotrophs using exogenic or artificial redox mediators [88], but also in lithotrophs, using inorganic electron donor such as H_2_, H_2_S or CH_4_, which are produced by geochemical reactions or the recycling of organic compounds. Systems promoting the maintenance and replication of these basic functions, such as RNA and DNA systems, would have been selected by evolution to form LUCA, a self-replicating and self-sufficient organism.

## 9. Electrotrophy, Serpentinization and Iron–Sulfur World

Once this hypothesis of the emergence of Life has been posed, it is necessary to examine its plausibility compared to other current hypotheses. Looking at the origin of Life in alkaline hydrothermal vents, we could consider that such natural electrical current is not possible in the modern carbonate chimney. Indeed, in the absence of a magmatic context, the alkaline hydrothermal fluid does not leach sulfur compounds that normally leads to the precipitation into the conductive pyrite [89]. Instead, CO_2_ precipitates into brucite, a non-conductive mineral and does not contain enough metallic particles to allow long distance electrical conduction [89]. However, during the Hadean, the chimney composition was assumed different, mainly composed of fougerite, also known as green rust, which is electrically conductive [59]. Then, we can hypothesize that an electrical current could have been present in ancient alkaline hydrothermal vent and could be an alternative energy source for the reactions involved in the AHV theory. In fact, as mentioned before, this scenario would allow a larger potential difference between anode and cathode than in acidic vents to perform the CO_2_ reduction into different carbon-products (Figure 1). Recent studies using microfluidic reactor replicating these conditions have shown that CO_2_ reduction might require a pH gradient and high pressure of H_2_ to obtain high concentration and to produce sufficiently low potential at the anode [86,87].

Considering the possibility of electrical current in both hydrothermal contexts, acidic/magmatic and alkaline/serpentinization, several points previously redhibitory need to be revised in a new light (Table 1).

The serpentinization reaction, at the source of the alkaline sources, produces mainly H_2,_ CH_4_, and acetate. As previously mentioned, these molecules would then have provided the energy necessary for the proto-metabolism. However, CH_4_ and H_2_, which are poorly soluble molecules, and acetate require diffusion to a specific point to feed catalytic reaction, most likely leading to mass transfer limitation, and reduction in reaction kinetics. In acid smokers (where no serpentinization reaction takes place), very little H_2_ is produced, whereas large amounts of H_2_S are released [65]. H_2_S, even if highly reactive, is considered important in many prebiotic syntheses and modern metabolism [90]. In our hypothesis, the oxidation of the electron donor (H_2_S or H_2_) takes place at a long distance from the reduction of the electron acceptor by the conduction of electrons over centimeters to meters through the conductive hydrothermal chimney. Thus, only a low concentration of H_2_S can react with the elements of the proto-cell, located much too far away, but produce the reducing power necessary for the production of the electric current. Moreover, the electron supply is intensive and continuous (but fluctuating in intensity) over the entire conductive surface of the hydrothermal vent, concentrating on the most catalytic points, presenting a wide range of conditions, and allowing a larger area of proto-cell development.

In the hypothesis of the origin of Life in alkaline hydrothermal springs, the transmembrane proton gradient is produced by gradient in a mineral matrix between the alkaline fluid and the slightly acidic sea water [59]. In this case, the proton gradient is passive and is not driven by the metabolism. In our hypothesis, the ionic gradient, produced by redox reactions, feeds the pH difference between the inner and outer space of the proto-cell, as explained above. Thus, in our hypothesis, a proton driving force is already established between the interior and the exterior, probably leading, by adaptive pressure, to the proto-cell taking advantage of this ionic current for the development of primitive pyrophosphatase pumps [91] leading further to the energetic mechanism of ATP Synthase.

Under the conditions of alkaline hydrothermal vents to allow further syntheses, the first organic molecules, such as acetate, would have been produced by Fischer–Tropsch type reactions in the hydrothermal fluid or indirectly by serpentinization reactions. These compounds would then have had to diffuse through the mineral surfaces to be adsorbed and concentrated, allowing it to react with each other to complexify and polymerize. In the context of the geo-electrochemical reduction of CO_2_, the reaction takes place on a mineral surface, mainly composed of polymetallic sulfides and clay, which are known to potentially adsorb organic molecules and favor catalysis [92]. The second aspect concerns the pH, a primordial factor in prebiotic synthesis reactions. It is important to recall that the emergence of Life in an acidic context was originally rejected because polymerization reactions, necessary for the formation of amino acids, are difficult at acidic pH [61]. However, this idea remains controversial. Indeed, recent studies suggest that neutral and alkaline aqueous environments might not be beneficial places for RNA phosphodiester bond but also of the aminoacyl-(t)RNA and peptide bonds formation [93,94]. Furthermore, many other chemical reactions that give rise to organic compounds are only favored, or even possible, in acidic environments [47]. Furthermore, in our hypothesis, the internal increase in the pH of the proto-cell by the electrochemical ion gradient would lead to pH values in the range from 3.5 to 11 and above, allowing all potential conditions for the polymerization of amino acids. The fluctuation of the current, caused by changes in the internal resistance of the chimney by environmental conditions (such as the temperature, precipitation of the chimney material, etc.), or changes in the catalytic activity of the chimney surface, would lead to a pH fluctuation between alkaline and slightly acidic, increasing the polymerization process and solubilization of compounds at different pH.

Indeed, the availability of macro- and micro-nutrients indispensable for the growth of modern microorganisms is also affected by the environmental conditions, and especially the pH. For example, phosphate, which is central to modern energy metabolisms and present in the environment mostly in apatite (a combination of calcium and phosphate), is solubilized at low pH. Then, when the internal pH of the proto-cells is low, due to a low electrical current, phosphate could solubilize and may have permitted incorporation into organic compounds as phosphate esters, followed by a second set of chemical reactions that initiated primitive metabolic pathways involving phosphate [95,96]. It was demonstrated that gradient of pH could allow the synthesis of pyrophosphate PPi in hydrothermal context, especially enhanced by electrochemical reactions [95,97].

Concerning the nitrogen source, researchers have shown the necessity of ammonium or ammonia for prebiotic amino acid synthesis and its role in the structure of RNA and DNA. In the AHV theory, nitrate is reduced to ammonia, and potentially hydrazine, in a 2D layered structure of green rust (fougerite [47]. In the electrotrophic theory, the electroreduction in nitrate would produce a high concentration of ammonium and NO localized at the same point as the catalytic point of reduction of CO_2_.

Finally, most prebiotic synthesis theories focus on the first steps of the transition from inorganic to organic, but little consideration is given on the transition from these specific energetic systems (diffused in minerals, for example) to an independent planktonic cellular system, as observed in LUCA or modern prokaryotes. As discussed earlier, the electrotrophic theory, once stabilized, is not bound to a specific mineral matrix or specific molecule as an energy source, but at any electrochemical reaction on a metallic catalyst, and can easily be translocated into a planktonic state.

## 10. Conclusions

In conclusion, the various hypotheses presented here are based and inspired on the observation of the complex modern electrotrophy. Indeed, the transition from prebiotic to modern biology is often under considered and, most of time, totally detached from each other, leading to unreasonable conditions without a perspective of transition to viable biology. In this review, we aimed to translate a modern metabolism, the electrotrophy, into prebiotic synthesis steps in biologically viable conditions. This tentative reconstruction of the process for the emergence of Life presents some similarity and increased probability compared to other common hypotheses. However, the conditions for such spontaneous electrical current remain hypothetical. It is, therefore, necessary to study in more detail the viability of these theories from a thermodynamic, phylogenetic or geochemical point of view to validate our postulates. Multidisciplinary approaches could also be considered to reconstruct these conditions in an acidic or alkaline hydrothermal context of the early ocean.

## Figures and Tables

**Figure 1 life-13-00356-f001:**
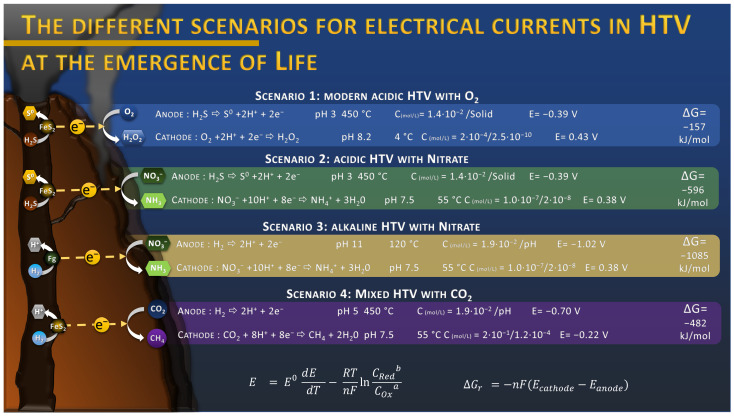
Estimation of onset potential E of half reactions and Gibbs free energy ΔG_r_ for the production of a spontaneous electrical current in different scenarios, depending on the chemistry of the HydroThermal Vent (HTV). Onset potential E were estimated with the Nernst equation, using the temperature coefficient dE/dT of each half reaction to correct the standard potential E^0^ [27], using average temperature estimated of the ocean during Archaean of 55 °C [29], and using concentrations C estimated in the literature for H_2_S [30], O_2_ [31], H_2_O_2_ [32], NO_3_^−^ [26], NH_4_^+^ [33], CO_2_ [30], and CH_4_ [34]. Concentrations are expressed as “reactant”/”product” concentrations. Potentials are expressed vs. SHE. All compounds are supposed to be dissolved in aqueous solution due to the high hydrostatic pressure (up to 600 bars), and the temperature coefficients are assumed to be constant from 25 to 450 °C, introducing a potential error. Fg: fougerite, FeS_2_: pyrite or associated polymetallic sulfur.

**Figure 2 life-13-00356-f002:**
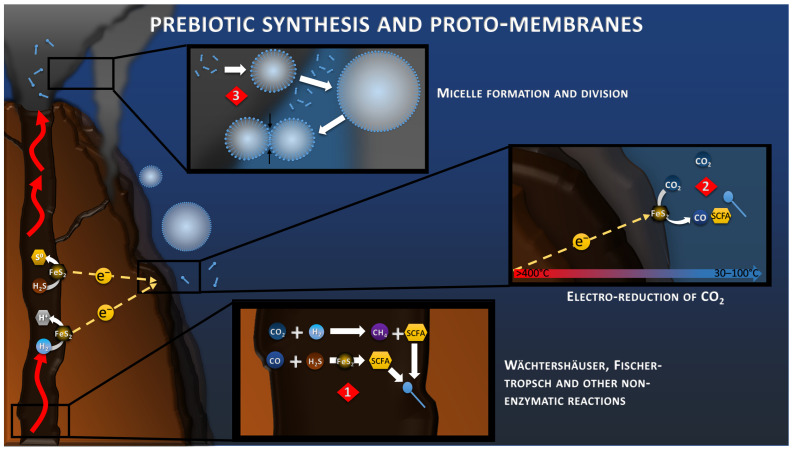
Wächtershäuser, Fischer–Tropsch reactions (reaction 1) and geoelectrochemical reduction of CO_2_ (reaction 2) toward the synthesis of Short Carbon Fatty Acids (SCFA). Concentration of SCFA would lead to the accumulation of amphiphilic molecules (blue pins), the formation of micelles, and the formation of the proto-membrane (reaction 3).

**Figure 3 life-13-00356-f003:**
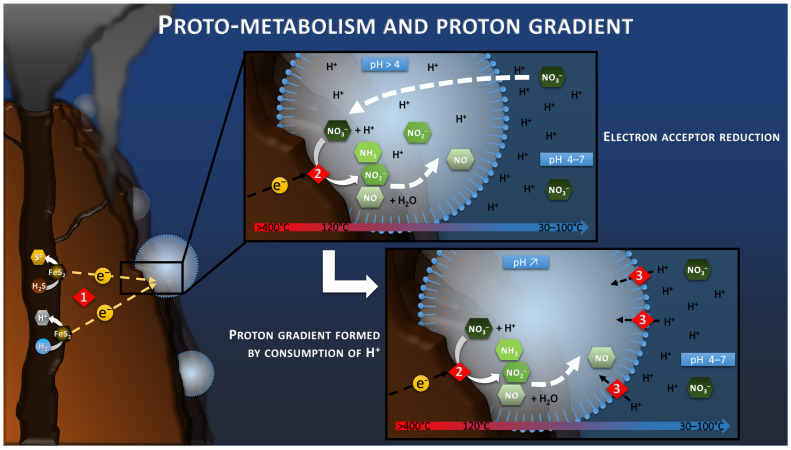
Schematic of the electrotrophy proto-metabolism hypothesis. Reaction 1 corresponds to the H_2_S or H_2_ oxidation step of the hydrothermal fluid leading to the production of the electric current and the release of protons. Reaction 2 corresponds to the reduction in nitrate by consuming the protons and electrons of the electric current. The confinement in a space bounded by the semi-permeable proto-membranes leads to the progressive depletion of proton from the intracellular space in case of nitrate reduction faster than the proton transport through the proto-membrane. This proton difference would lead to the creation of a driving proton force (Reaction 3).

**Figure 4 life-13-00356-f004:**
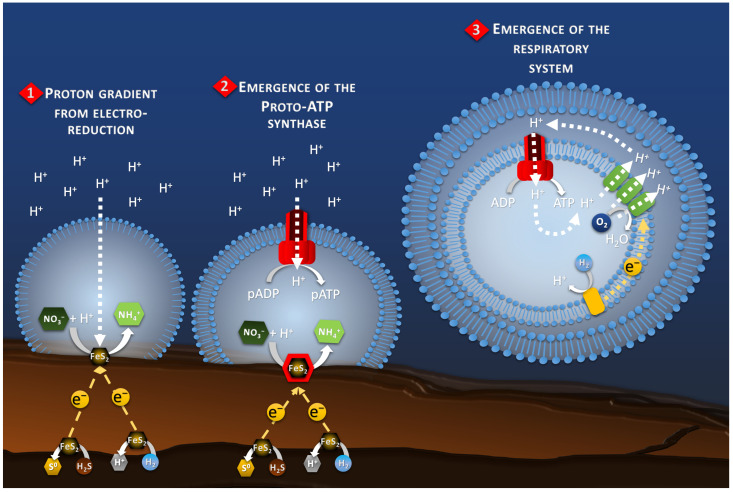
Schematic of the hypothetical evolution of the proton-motive-force system. It starts, in the Part 1, with the natural proton gradient produced by the electro-reduction reactions on the chimney wall. The Part 2 represents the transition to a proto-ATP synthase transforming proto-ADP (pADP) into proto-ATP (pATP). Finally, the Part 3 shows the development of a proto-respiratory chain system (green blocks) fed by a proto-enzyme oxidizing an electron donor (yellow block), for example H_2_, to maintain the proton gradient while detaching from the chimney wall and the electrical current as energy source.

**Figure 5 life-13-00356-f005:**
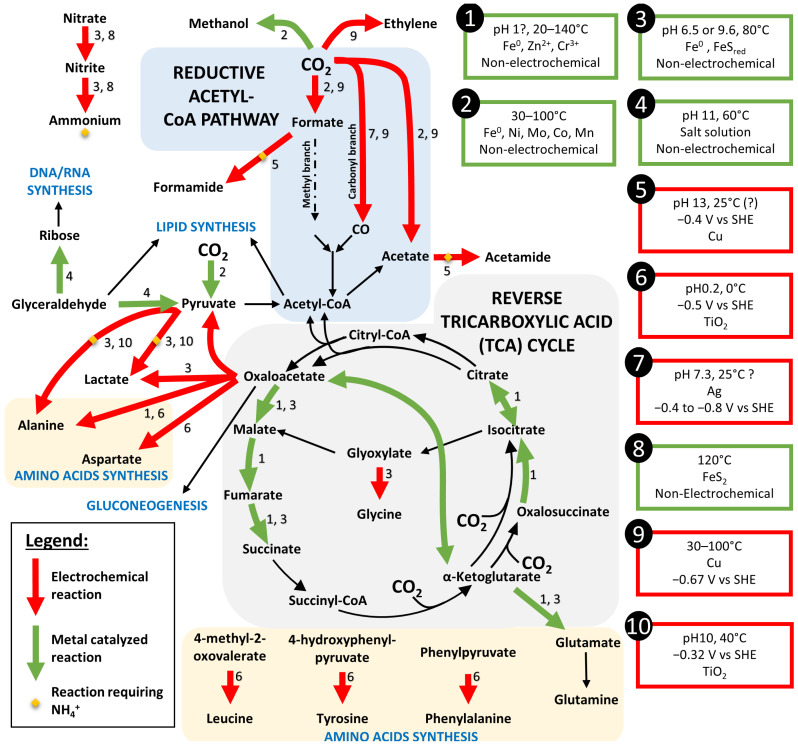
Reconstitution of the autotrophic pathways through prebiotic synthesis by metal catalyze (green arrows) or electrochemical reaction (red arrows) from the literature review. When both type of reaction were possible, the arrow was set in red. Black arrows represent the steps of the pathways not yet replicated by prebiotic synthesis. Conditions of each reaction is referred in the boxes. References: 1—[71], 2—[68], 3—[69], 4—[72], 5—[73], 6—[74], 7—[75], 8—[62], 9—[76], 10—[74].

**Figure 6 life-13-00356-f006:**
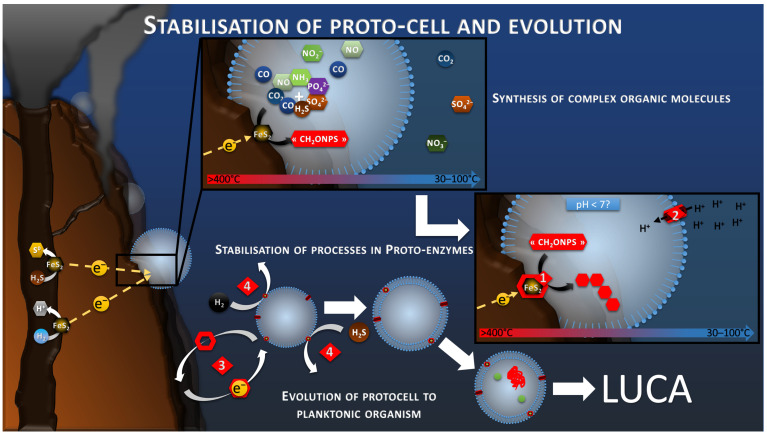
Schematic of the development of autotrophic and anabolic proto-metabolisms, leading to polymerization (reaction 1), the stabilization of the processes in an organic matrix, and to proto-enzymes (reaction 2). Stabilized processes and proto-membrane allow the translocation on the proto-cell in planktonic phase using redox mediators (reaction 3), diversification of the electron donor to soluble compounds (reaction 4), and the evolution to a self-replicative system.

**Table 1 life-13-00356-t001:** Comparison of different factors for the emergence of Life in the three hypotheses in hydrothermal context.

Critical Aspects	Iron–Sulfur World	Alkaline Hydrothermal Vent	Electrotrophy
Driving process	Magmatic hydrothermalism	Serpentinization	Electrical current from hydrothermalism
Energy source	H_2_S Too reactive to allow polymerization?	H_2_ or CH_4_ Low diffusion across mineral structure	Direct electrons from electricity Constant or fluctuating, concentrated at a specific catalytic point
Electron sink	Not defined	Nitrate	NO, Nitrate, Sulfate, CO_2_
Carbon fixation	High solubility at low pH Reaction with H_2_S catalyzed on FeS_2_	Precipitation into carbonate Low solubility of aqueous carbon Catalyzed by fougerite or during serpentinization reactions	High solubility at low and neutral pH CO_2_ electroreduction catalyzed by metals
Localization	Diffused in the hydrothermal fluid and all chimney surrounding	Localized in the chimney structure	Localized on all conductive surface of the chimney and surrounding
Proton gradient	From mixing of acidic hydrothermal fluid (pH~3) with neutral seawater (pH~6–7)	From mixing of alkaline hydrothermal fluid (pH~11) with neutral seawater (pH~5–6)	From H^+^ consumption during electroreduction, fluctuating with current
Availability of Phosphorous	High solubility at low pH	Low solubility at high pH	Fluctuating solubility with changing pH due to fluctuating current
Availability of Nitrogen	NO reduced to NH_4_^+^ by H_2_S or metals	Reduction of NO_3_^−^ to NH_3_^+^ in fougerite	Electroreduction of NO_3_^−^, NO_2_^−^ to NH_4_^+^ on pyrite
Heavy metals	Present in the chimney	Mo and W soluble at high pH	Present in the catalytic chimney
Temperature	Very high temperature from hydrothermal fluid (up to 400 °C) Unstable biomolecules	Intermediate temperature from serpentinization (up to 120 °C)	Intermediate to low temperature on the surface of the chimneys, mixed with seawater
Amino acids formation	Wächtershäuser’s reaction by FeS, CO_2_ and H_2_S	Catalyzed by iron oxyhydroxide matrix	Electrosynthesis from α-keto acids and metal catalyst
Polymerization	Low polymerization at low pH and high temperature Condensation on pyrite	Higher polymerization at high pH and moderate temperature Catalyzed by fougerite	Electropolymerization? Condensation on pyrite Catalyzed by metals
Translocation of the energetic system	Dependent on the hydrothermal fluid	Dependent on the chimney structure and serpentinization reactions	Easily translocated into planktonic state using redox mediators

## Data Availability

Not applicable.

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
