# Peer review of "Spark of Life: Role of Electrotrophy in the Emergence of Life"

_life, 2023, doi:10.3390/life13020356_

Round 1

Reviewer 1 Report

In this informative and meticulous manuscript, the authors delineate recent discoveries in electrotrophy, and then utilize these findings to construct a plausible path for the origin of life at submarine hydrothermal vents. I found this work to be interesting and quite compelling, but there are a number of minor lacunae (described below) that must be addressed before the manuscript can be deemed suitable for publication.

1. One of the first and major comments I have is that the authors should explicitly list some of the candidates (other than submarine hydrothermal vents) that have been proposed as sites of abiogenesis in Section 2. Notable examples include aerosols, hot springs (on land), hydrothermal sediments, beaches, lagoons, and more. Mentioning these environments will give the reader a broader picture of what habitats might have been conducive for the origin(s) of life beyond submarine hydrothermal vents; otherwise, the readers will mistakenly think that hydrothermal vents are the only candidates discussed so far in the scientific literature. References that can be cited in this context are as follows:
https://onlinelibrary.wiley.com/doi/full/10.1111/gbi.12025
https://www.liebertpub.com/doi/full/10.1089/ast.2019.2045
https://www.hup.harvard.edu/catalog.php?isbn=9780674987579

2. Ref. [7] cited in lines 55-56 on pg. 2 is a bit shaky because the precipitates are not confirmed to be genuine microfossils. Hence, the authors should change their wording slightly from "bacterial fossils" to "putative microfossils" to reflect the uncertainty.

3. When the authors discuss the AHV theory in lines 75-76 of pg. 2, they are recommended cite the following early papers by Russell and collaborators, which set the stage for later publications.
https://link.springer.com/article/10.1007/BF00160147
https://onlinelibrary.wiley.com/doi/abs/10.1111/j.1365-3121.1989.tb00364.x

4. In connection with line 85 of pg. 2, it is worth mentioning the resistivity of a metal (e.g., copper) vs an insulator (e.g., wood), so that the readers can see that the resistivity lies somewhere in between.

5. Ref. [19] cited in the paper a few times has been contested by subsequent papers (Ranjan et al. 2019). Hence, the authors should acknowledge that the reported theoretical models are tentative.
https://agupubs.onlinelibrary.wiley.com/doi/full/10.1029/2018GC008082
The emphasis on nitrate at later stages should also be modified slightly to reflect the fact that we do not have clear constraints on the abundance of this species in the Hadean ocean.

6. Figure 1 is helpful, and I like the layout. However, since the journal has a broad audience, it is worth explicitly indicating how Delta G is calculated from the redox potentials, namely Delta G = n * F * Delta E, where "n" is the no. of electrons transferred.

7. Line 143 on pg. 4 is slightly confusing in its current form. The authors should clarify further the differences between 590 mV and 300 mV cases. Moreover, since CO2 is the only carbon source, it may be worth remarking that the pathway is autotrophic in nature.

8. In lines 196-199 (on pg. 5), the authors mostly focus on Fischer-Tropsch type reactions, but several other prebiotic compounds of interest have been synthesized in conditions resembling alkaline hydrothermal vents - a recent review of the subject is furnished below.
https://pubs.acs.org/doi/full/10.1021/acs.chemrev.0c00191
This aspect is worth mentioning in the paper, both on pg. 5 and later.

9. Section 5 on pg. 5 is a big paragraph that behaves like a wall of text making it difficult to parse the relevant details. Please consider breaking it down into smaller paragraphs, and perhaps creating a new table with the salient values of the redox potentials.

10. In lines 237-238 on pg. 6, it would be helpful if the authors can clarify further what type of micelles are formed in lab experiments: monolayer vs bilayer, regular vs inverted structure.

11. In the context of lines 264-272 on pg. 7, it is worth mentioning that hydrocarbons can be synthesized indirectly via serpentinization, as shown in the experiments performed by McCollom and colleagues.

12. In line 311 on pg. 8, I suggest reminding the readers about what the "cathodic reaction" refers to (presumably the reduction of oxidants).

13. In Section 8, there is a distinct emphasis on the Wood-Ljungdhal (WL) pathway, but there is less attention devoted to the reverse Krebs cycle, which has witnessed many promising developments in origin-of-life studies (see the citation in my point #8). The authors should think about extending the discussion of the reverse Krebs cycle.

14. I wish to compliment the authors on their Figure 5. It is rather dense, but it conveys a lot of useful information in a compact manner. It is a bit blurry so the authors should replace it with a high-res version when the manuscript is accepted for publication.

15. An issue with the pathway in lines 427-428 (on pg. 12) is that it requires hydroxylamine, and the concentration of this molecule in the Hadean ocean is not well understood; this caveat should be specified somewhere.

16. Some limitations should be underscored regarding Section 8.4, because most studies on clay minerals and polymerization have not been undertaken in conditions analogous to hydrothermal vents. For instance, if the authors peruse the recent review of Ref. [72], these environments are explicitly mentioned only a couple of times.

17. I feel that Section 9 is too brief, and does not provide a great deal of information concerning the origin of life - instead, it is a brief and speculative glimpse into early life. I would suggest removing this section altogether (or expanding it), because it breaks the flow of the rest of the manuscript.

18. Table 1 was clearly constructed and represents a valuable summary of the electrotrophy hypothesis. In light of my previous comments, however, please make any revisions or updates to the table if they are needed.

19. Page 15 (Section 10) contains several points that were already made in the previous sections such as the reactivity of hydrogen sulfide. Hence, please go through this section once again and make sure that such content is reduced or suitably reframed.

20. In connection with the origin-of-life at acidic pH (lines 542-545), the following reference seems worth citing.
https://biologydirect.biomedcentral.com/articles/10.1186/1745-6150-7-4

21. I like the current title, but the authors might consider modifying it to "...electrotrophy in facilitating the emergence..."

Reviewer 2 Report

The manuscript “Spark of Life: Role of electrotrophy in the emergence of Life” leaves a strange impression with me. It reads like a review, but lacks over wide areas of the text any experimental results. So it is more like a proposal for experiments, but then does not instruct us which experiments should be done. Where it becomes really aggravating is when the authors lack to cite experiments that were possibly the closest to their ideas: PNAS 117, 22873-22879 (2020), doi.org/10.1073/pnas.2002659117.

So I fear the authors are not really up to the task and the paper is just another highly speculative manuscript on the Origins of Life which do our field a big disfavor: colorful images over experimental results, insisting on hypothesis although experiments have disproved them already by not citing them or showing that it is a problem of the experiment (page 13, bottom). It is the experiment that is the source of scientific truth, not the hypothesis!

I understand that elife is not the most prestiguous journal outlet, but how much should I be forced to tolerate? As experimentalist working in the field, I find such theory ladden papers very annoying since the ignore even the most simple physical and chemical facts. I could accept the manuscript if at least the authors could provide enough expertise to make a Comsol simulation of their hypothesis to for example show how fast the kinetics of an assumed initial non-equiliibrium relaxes to the equilibrium state. Would it be microseconds or milliseconds? So, no, the shown experimental ideas would not even make it through our first filter of creating an experiment, let alone make it into a paper since I believe that experiments, not hypothesis is driving this field – as it does in all other fields of science.

Random points that come to my mind:

1) Missing discussion of microfluidic approaches to test the presented hypothesis

2) Missing discussion that and how non-equilibrium should be maintained in their colorful images. Most of them involve membranes which is known to speed up the local thermodynamic equilibration. So even if the experiments could be realized, I am missing the point how the electrical fields could be sustained at the small scale against the fast diffusional relaxation.

3) The authors neglect the fact that more and more papers are discussing the role of CO on early Earth atmospheres and Mars. They seem to write this manuscript without keeping track of the most recent developments, just replicating with images what Michael Russell – not very much an experimentalist – and others have continued to bore us with in meetings.

So to write it with the words of the authors: “In conclusion, the various hypotheses presented here are based and inspired on the observation of the complex modern electrotrophy”. Which observation? I should oppose the publication of this manuscript since it will lay down utterly wrong tracks for future experimentation and does not even provide an overview over the literature. In other fields such a manuscript would be directly rejected by the editor. I leave it up to him/her to decide.

Round 2

Reviewer 1 Report

The authors have done a thorough job of addressing my comments and incorporating my feedback. Hence, while I recommend the paper for publication, the following minor edits should be implemented first:

1. In Figure 2, the label "Wächtershäuser and Fischer-Tropsch reactions" should be reversed to read "Fischer-Tropsch and Wächtershäuser reactions" because the top reaction is Fischer-Tropsch-type and bottom one is Wächtershäuser-type.

2. Lines 284 and 285 are actually two sides of the same coin, because Fischer-Tropsch-type reactions reliant on molecular hydrogen produced during serpentinization are capable of producing methane and other hydrocarbons. Hence, minor rewriting of lines 283-287 is suggested, and Refs. 17, 57-59 can be cited together instead of separately.

3. The first time in the paper where the term "Energy source" is introduced, the authors should add in parentheses that this term could be synonymous with reducing agent (i.e., electron donor).

Author Response

The authors have done a thorough job of addressing my comments and incorporating my feedback. Hence, while I recommend the paper for publication, the following minor edits should be implemented first:

  1. In Figure 2, the label "Wächtershäuser and Fischer-Tropsch reactions" should be reversed to read "Fischer-Tropsch and Wächtershäuser reactions" because the top reaction is Fischer-Tropsch-type and bottom one is Wächtershäuser-type.

The label has been changed accordingly

  1. Lines 284 and 285 are actually two sides of the same coin, because Fischer-Tropsch-type reactions reliant on molecular hydrogen produced during serpentinization are capable of producing methane and other hydrocarbons. Hence, minor rewriting of lines 283-287 is suggested, and Refs. 17, 57-59 can be cited together instead of separately.

Lines 283-287 have been changed accordingly

  1. The first time in the paper where the term "Energy source" is introduced, the authors should add in parentheses that this term could be synonymous with reducing agent (i.e., electron donor).

“(electron donor)” have been added to line 25 and 76

Reviewer 2 Report

I am fine with accepting the revised manuscript. Still I like more experimentally founded work.

Author Response

Thank you